# Giant Cell Tumor of Bone in Patients under 16 Years Old: A Single-Institution Case Series

**DOI:** 10.3390/cancers13112585

**Published:** 2021-05-25

**Authors:** Francesca Ambrosi, Alberto Righi, Stefania Benini, Giovanna Magagnoli, Ilaria Chiaramonte, Marco Manfrini, Alessandro Gasbarrini, Tommaso Frisoni, Marco Gambarotti

**Affiliations:** 1Department of Pathology, IRCCS Istituto Ortopedico Rizzoli, 40136 Bologna, Italy; fra.ambrosi@gmail.com (F.A.); alberto.righi@ior.it (A.R.); stefania.benini@ior.it (S.B.); giovanna.magagnoli@ior.it (G.M.); 2Department of Orthopaedic Oncology, IRCCS Istituto Ortopedico Rizzoli, 40136 Bologna, Italy; ilaria.chiaramonte@ior.it (I.C.); marco.manfrini@ior.it (M.M.); tommaso.frisoni@ior.it (T.F.); 3Department of Oncologic and Degenerative Spine Surgery, IRCCS Istituto Ortopedico Rizzoli, 40136 Bologna, Italy; alessandro.gasbarrini@ior.it

**Keywords:** giant cell tumor, bone, pediatric, H3F3A, immunohistochemistry

## Abstract

**Simple Summary:**

Giant cell tumor of the bone is a locally aggressive, rarely metastasizing tumor that accounts for about 5% of bone tumors; it generally occurs in patients between 20 and 45 years old. Sporadic cases (less than 140) have been described as occurring in the first two decades of life. A histone 3.3 (H3.3) gene, *H3F3A*, has been recently identified in as many as 96% of giant cell tumors of bone. These mutations are useful in the differential diagnosis of giant cell tumor of bone with its mimickers. The immunohistochemical expression of H3F3A resulted comparable to molecular analysis as reported in a recent investigation. In the present study, we describe our series of giant cell tumors of bone in pediatric patients <16 years old.

**Abstract:**

Background: Giant cell tumor of bone is a locally aggressive, rarely metastasizing tumor that accounts for about 5% of bone tumors and generally occurs in patients between 20 and 45 years old. A driver mutation in the histone 3.3 (H3.3) gene *H3F3A* has been identified in as many as 96% of giant cell tumors of bone. The immunohistochemical expression of H3F3A H3.3 G34 expression was found in 97.8% of cases. In the present study, we describe our series of cases of giant cell tumor of bone in pediatric patients <16 years old. Methods: All cases of giant cell tumor of bone in pediatric patients <16 years old treated in our institute between 1982 and 2018 were reviewed. Immunohistochemistry and/or molecular analysis for *H3F3A* gene mutations was performed to confirm the diagnosis. A group of aneurysmal bone cysts in patients <16 years old was used as a control group. Results: Fifteen cases were retrieved. A pronounced female predominance (93%) was observed. A pure metaphyseal central location occurs in 2 skeletally immature patients. Conclusions: Giant cell tumor of bone should be distinguished from its mimickers due to differences in prognosis and treatment. Immunohistochemical and molecular detection of *H3F3A* gene mutation represents a reliable diagnostic tool.

## 1. Introduction

Giant cell tumor of the bone (GCTB) is a locally aggressive, rarely metastasizing tumor composed of neoplastic mononuclear stromal cells, with macrophages and multinucleated reactive giant cells (osteoclast-like) uniformly distributed [1,2,3]. GCTB accounts for about 5% of bone tumors, and it occurs mostly in skeletally mature patients, generally between the ages of 20 and 45 years [3]. However, sporadic GCTB has been reported in older patients, as well as in the first two decades of life [4,5,6,7,8,9], in which less than 140 cases have been reported. According to the last WHO Classification of Soft Tissue and Bone tumors, GCTB has been included in the group of osteoclastic giant-cell-rich tumors [3].

In adult patients, GCTB is generally located in the meta-epiphysial region of the long bones, eccentrically, especially the distal femur, proximal tibia, and distal radius; spine, sacrum, and pelvis can also be affected. It is rare in short tubular bones of the hands and feet [5]. In pediatric patients is often located in the metaphysis [10]. Multifocal metachronous and/or synchronous cases are rare; in addition, giant-cell-rich lesions histologically similar to giant cell tumors of bone are observed in a specific subset of patients with Paget’s disease [11,12,13].

Radiologically, GCTB has characteristic and diagnostic features, especially in adult patients and in common sites. It appears as a well-defined, bordered, eccentric, lytic, subchondral lesion that involves epiphysis and metaphysis with the typical “soap bubble” appearance, while the bone cortex is expanded and could be focally destroyed. Usually, borders do not show margins of sclerosis or trabeculation [14].

Histologically, mononuclear round to oval and spindle-shaped cells are dispersed together with multinucleated reactive osteoclast-like giant cells, evenly distributed; multinucleated giant cells have a variable number of nuclei, also more than 50 per cell. The mitotic rate in the mononuclear cells can be quite high, but no atypical forms are seen. The classic setting of GCTB can be modified by a secondary reactive proliferation of fibro-histiocytic tissue, areas of hemorrhage, necrosis, and secondary aneurysmal bone cyst-like features, and a different growth pattern has been described [3,4].

The main differential diagnosis of GCTB is with aneurysmal bone cyst (ABC) with “solid” areas (previously called giant cell reparative granuloma of small bones), other benign osteoclastic giant-cell-rich tumors such as a brown tumor of hyperparathyroidism and non-ossifying fibroma, and malignant bone tumors rich in reactive osteoclastic giant cells [15,16,17,18]. In this setting, a driver mutation in the histone 3.3 (H3.3) gene *H3F3A* has been recently identified in as many as 96% of GCTB cases [3,19]. Although these mutations can be present in sarcomas secondary to GCTB [20], they are useful in the differential diagnosis of GCTB and its mimickers, especially in borderline cases without clear-cut clinical and radiological context [21]. The immunohistochemical expression of H3F3A resulted comparable to molecular analysis as reported in recent investigations where H3.3 G34 expression was found in 97.8% of GCTB [20,21,22].

Although GCTB is considered a tumor that affects skeletally mature patients, it has been rarely reported in skeletally immature patients [4,5,6,7,8,9,21,22]. A recent series reviewed 63 patients under 18 years old, and only 5 were GCTB patients harboring *H3F3A* gene mutations, confirmed by PCR analysis [4].

The present study aims to describe the cases of GCTB in patients <16 years old treated in our institute. All clinical, radiological, and histological data were reviewed.

## 2. Materials and Methods

We retrieved all cases of GCTB in patients <16 years old from the archive of the Rizzoli Orthopedic Institute, treated between 1982 and 2018. Cases of brown tumor of hyperparathyroidism and non-ossifying fibroma were excluded due to the peculiar clinical–serological or radio-histological features, respectively.

Inclusion criteria were as follows:Age <16 years old;Sufficient clinical information available;Histological slides and histological material suitable for immunohistochemistry or genetic analysis for *H3F3A* gene mutations available.

In all selected cases, all histological, radiological, and clinical features were reviewed.

Immunohistochemistry analysis for H3F3A was performed to confirm the diagnosis on undecalcified tissue. This was evaluated on paraffin-embedded tumor specimens with anti-histone H3.3; we used the primary antibodies against histone H3.3 G34W mutant protein (rabbit monoclonal, clone RM263, dilution 1:600; RevMAb Biosciences, South San Francisco, CA, USA), histone H3.3 G34R mutant protein (rabbit monoclonal, clone RM240, dilution 1:500; RevMAb Biosciences), and histone H3.3 G34V mutant protein (rabbit monoclonal, clone RM307, dilution 1:500; RevMAb Biosciences) [22]. Sections from the paraffin tumor blocks were cut with a microtome and mounted on microscope slides for immunohistochemical analysis. Unstained sections (tumor and control) were heat-treated at 60 °C for 20 min, deparaffinized, and immunostained on a Ventana BenchMark following the manufacturer’s guidelines (Ventana Medical Systems, Tucson, AZ, USA). Standard avidin–biotin complex peroxidase assays were performed to analyze the expression of H3F3A. Adult GCTB with molecular confirmation of *H3F3A* gene mutation was used as a positive external control. Based on the previously published study that evaluated the nuclear expression of H3F3A, it was considered positive with unequivocal strong crisp nuclear expression and negative with a not-detected expression [20,21,22,23].

In cases negative after immunohistochemistry, Sanger sequencing analysis for *H3F3A* gene variation was performed. For this analysis, >60 ng of DNA was amplified using AmpliTaq Gold 360 Master Mix (Applied Biosystems, Foster City, CA, USA) with 0.5 μM of primers (H3F3A: forward TGT TTG GTA GTT GCA TAT GGT GA; reverse H3F3A3 ACA AGA GAG ACT TTG TCC CAT T 239 bp [24]). The sequencing was performed by Bio-Research Fab (http://www.biofabresearch.it (accessed on 31 March 2021) Rome, Italy). Mutation analysis was conducted with Basic Local Alignment Search Tool (BLAST) in the NCBI database “National Center of Biotechnology Information Database” (http://www.ncbi.nlm.nih.gov/BLAST (accessed on 31 March 2021)). Electropherograms were exported to fast format and were aligned to the NCBI BLAST sequence of NM_002107.4 H3F3A mRNA. The samples negative to the sequencing analysis with the Sanger method were subsequently tested with allele specific locked nucleic acid quantitative PCR (ASLNAqPCR) as described in a previous paper [25]. A control group of ABC in patients <16 years old morphologically suspicious for the diagnosis of GCTB was used as a control group and analyzed immunohistochemically and genetically for H3F3A gene mutation.

Informed consent was collected from all patients by the standard procedure and with Rizzoli Institute ethics committee approval (CE AVEC 377/2019/Oss/IOR, 7 June 2019).

## 3. Results

From 1982 to 2018, a total of 910 cases of GCTB were retrieved from the archives of the Rizzoli Orthopedic Institute.

Nineteen cases met the inclusion criteria. Of note, eight excluded cases were originally diagnosed as GCTB in patients <16 years old but lacked histological material or sufficient documentation (counseling cases).

These 19 cases were tested immunohistochemically for H3.3 p.Gly34Trp (G34W, G34R, G34V). Fourteen cases were positive for G34W antibody and included in the study (Figure 1A). The five negative cases were subsequently molecularly analyzed for H3F3A gene mutation: three were negative, one was not evaluable, while the last one (case 11, Table 1) was positive (Figure 1B); only this last case was added to the series, reaching 15 cases. Considering that Sanger sequencing is the most used method with a limitation due to its limit of detection of about 20%, the three negative samples were subsequently confirmed also with ASLNAqPCR [26]. The other not-confirmed four cases were excluded from the study.

The control group consisted of 27 ABCs in patients <16 years old, morphologically suspicious for the diagnosis of GCTB. All these cases were negative both immunohistochemically and genetically for H3F3A gene mutation.

Clinical, radiological, histological, immunohistochemical, and molecular data of these selected 15 patients are summarized in Table 1.

Fourteen patients were female and only one was male. The mean and median age was 13 years old (range 8–15). In 5 cases (7, 8, 10, 11, and 14) initial imaging was not available and the location was only described in the radiological reports. In cases with available imaging, in long bones, two cases (cases 1 and 2) occurring in skeletally immature patients with an open growth plate were purely metaphyseal and centrally located (Figure 2A), while the seven cases occurring in skeletally mature patients with a closed growth plate were meta-epiphyseal and eccentric (case 15, Figure 2B).

Histologically, all cases showed typical histological features of GCTB (Figure 2C,D).

One patient was treated with denosumab and was alive with disease at the last follow-up. All the other patients were treated with curettage or resection, and two of these patients (cases 1 and 3) suffered from local recurrences treated with a new curettage; case 3 was lost at follow-up, while case 1 was alive without disease at the last follow-up.

## 4. Discussion

GCTB accounts for about 5% of bone tumors. Its common peak incidence is between the ages of 20 and 45 years. It has rarely been reported in pediatric patients, and a principal case series review of English literature is analyzed, compared to our experience, and shown in Table 2 [4,5,6,7,8,9]. The paper of Picci et al. [9] reported six cases treated at the Rizzoli Orthopedic Institute; however, these patients were treated before 1982, and for this reason, they are not included in the present series.

In the present study, we describe a series of 15 cases of GCTB in pediatric patients <16 years old, treated from 1982 to 2018 at the Rizzoli Orthopedic Institute.

As previously reported by other authors, GCTB in pediatric patients has a rare incidence (15/910, 1%) [4,5,6,7,8,9].

In our series, there is a pronounced female predominance (93%), while in the literature only a slight female predominance is reported [3], consistently noted in pediatric series [4,5,6,7,8,9]. Picci et al. [9] also reported a pronounced female predominance.

In our series, 87% of cases arose in long bones (tibia 8/15, 53%; femur 3/15, 20%; and radius 2/15, 13%), confirming previous reports [3,4,5,6,7], while one case each arose in sacrum and vertebra (7% each). No cases arose in the small bones of the hands and feet or the skull or showed a multifocal presentation.

In adult patients, the typical location of GCTB is in the meta-epiphyseal region of long bones, eccentrically. In our series, considering the cases occurring in long bones with available imaging (nine cases), in the two skeletally immature patients with an open growth plate (case 1 and 2), the lesions were purely metaphyseal and centrally located, while in the seven patients with a closed growth plate, the lesions were meta-epiphyseal and eccentric. This does not exactly correspond to the description of Campanacci [10] of a prevalent pure metaphyseal location in pediatric patients. However, if we consider the skeletal age of the patient and the status of the growth plate, the fact that the lesions showed a pure metaphyseal location in the two skeletally immature patients supports the hypothesis of a metaphyseal origin of GCTB. In the literature, it is difficult to check the exact correspondence between the status of the growth plate and the location of the lesions, since this is not described in each series. Picci et al. [9] described six cases of GCTB in skeletally immature patients with an open growth plate; with the limitation that these cases were not tested for the presence of H3F3A mutation, all of them showed a predominant metaphyseal location, with extension into the epiphysis in five patients, thus demonstrating that the presence of an open growth plate does not preclude the possibility of an epiphyseal extension, beyond the physis.

It is important to differentiate GCTB from its mimickers, particularly from ABC with “solid” features that can show similar clinical, radiological, and histological features and that is more frequent in pediatric patients.

Immunohistochemistry for H3.3 p.Gly34Trp (G34W, G34R, G34V) is a useful and predictable surrogate marker for molecular analysis in distinguishing GCTB from its mimickers.

Although clinically most of the cases of GCTB and other osteoclastic giant-cell-rich lesions behave in a similar benign fashion and have a similar treatment, patients with GCTB can rarely develop lung metastasis (3–7% of cases) or a secondary malignant transformation, mostly after radiation therapy [3,15,16,17,18,27], thus indicating the importance of differentiating between these entities for specific clinical management, especially in pediatric patients who could be affected by growth-related problems and angular deformities after surgery. This is the reason why in the last WHO classification [3], ABC is classified as a benign tumor, while GCTB is classified as locally aggressive, rarely metastasizing neoplasm.

## 5. Conclusions

In conclusion, we describe our series of GCTB in pediatric patients <16 years old. Although rare (1% of all GCTB), they can occur also in these age groups, with a pronounced female predominance (93%). Differently from the typical meta-epiphyseal location, in skeletally immature patients with an open growth plate, GCTB can be limited to the metaphysis or have a predominant metaphyseal location, with extension into the epiphysis. GCTB should be distinguished from other osteoclastic giant-cell-rich tumors, because of the differences in prognosis and treatment. Immunohistochemical or molecular detection of *H3F3A* gene mutation represents a specific, accessible, and reliable diagnostic tool in the differential diagnosis with other mimickers and pathologists should be aware to use these techniques in doubtful cases, also in young patients.

## Figures and Tables

**Figure 1 cancers-13-02585-f001:**
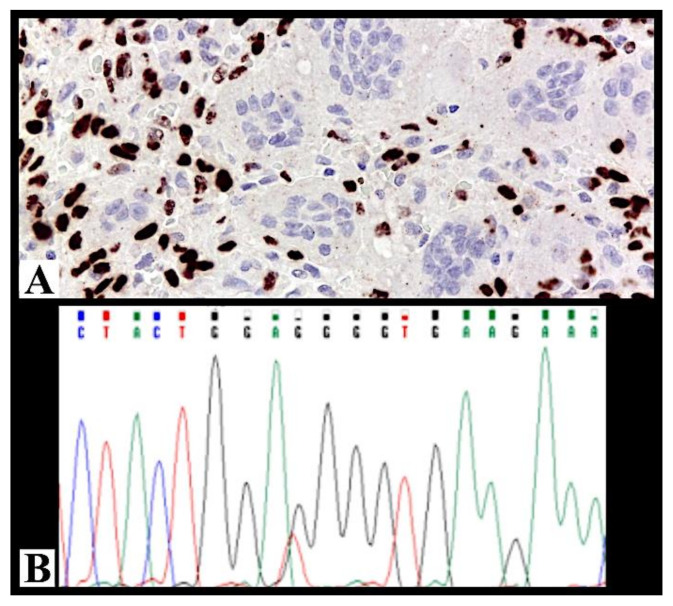
(**A**) Strong and diffuse immunohistochemical nuclear positivity for H3F3A in case 10 (immunohistochemistry for H3F3A, 200× magnification). (**B**) Molecular analysis of case 11 confirmed the presence of the mutation in heterozygosity.

**Figure 2 cancers-13-02585-f002:**
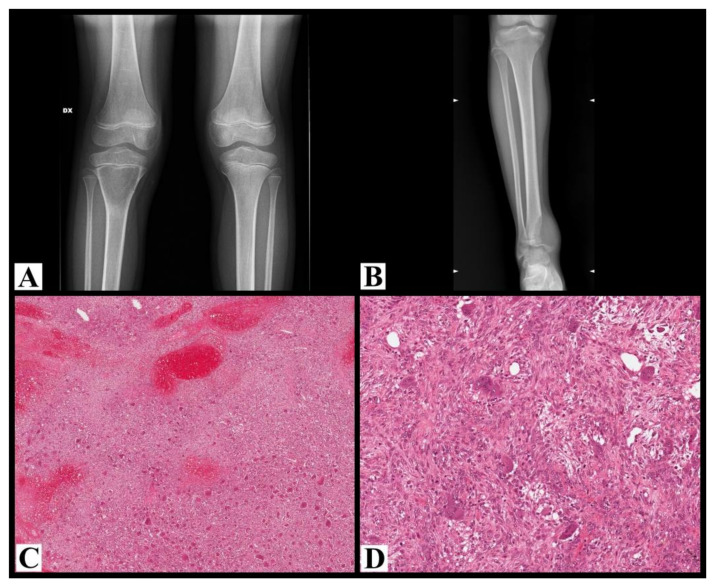
(**A**) Plane X-ray of the knee shows a purely lytic, centrally located lesion in the proximal metaphysis of the tibia with relatively well-defined margins underneath an immature growth plate; the cortex is thinned without interruptions (case 1). (**B**) Plane X-ray of the ankle shows osteolysis of the distal tibia located in the meta-epiphyseal region with aggressive radiological features: not well-defined margins, cortex expanded and interrupted, growth plate invaded and crossed (case 15). (**C**) Histological examples of giant cell tumor of the bone (case 14), with a highly vascular stroma, fibrosis, and reactive woven bone; acute hemorrhage, hemosiderin, xanthomatous histiocytes are admixed with neoplastic cells (hematoxylin and eosin, 50× magnification). (**D**) At higher magnification, the same tumor is composed of numerous osteoclast-like giant cells uniformly distributed throughout the tumor, which are larger than normal osteoclasts with numerous (>30) nuclei; mononuclear round to oval and spindle-shaped cells are dispersed together with the giant cells (hematoxylin and eosin, 200× magnification).

**Table 1 cancers-13-02585-t001:** Clinical, radiological, histological, immunohistochemical, and molecular features of the 15 patients are collected in the table.

N°	Sex	Age	Site	Status of Growth Plate	Size (Major Axis)	IHC H3F3A G34W	IHC H3F3A G34R	IHC H3F3A G34V	Molecular Analysis	Surgery	LR	Complications	Last FU (Months)	Status
1	F	8	Tibia, proximal central metaphysis	open	5	+	-	-	na	curettage and cement (+bone grafts)	yes—8 months	Leg length discrepancy	48	ned
2	F	10	Femur, left central metaphysis	open	6	+	-	-	na	chondrodiastasis and intercalar	NO	none	112	ned
3	F	11	Tibia, eccentric distal, meta-epiphysis	closed	4	+	-	-	na	curettage and cement (+bone grafts)	NO	none	lost	na
4	F	12	Tibia, eccentric distal, meta-epiphysis	closed	4	+	-	-	na	curettage and bone grafts	NO	none	123	ned
5	F	13	Femur, distal meta-epiphysis	closed	9	+	-	-	na	curettage and cement	NO	none	90	ned
6	F	13	Tibia, distal meta-epiphysis,	closed	5	+	-	-	na	massive grafting	NO	graft obturation, revision with fibula	42	ned
7	F	13	Radius, distal, meta-epiphysis	na	na	+	-	-	na	osteoarticular resection and grafting	NO	graft failure, revision in arthrodesis	204	ned
8	F	13	Tibia, central proximal metaphysis	na	na	+	-	-	na	na	yes—12 months	none	14	ned
9	F	14	Sacrum	indeterminate	8	+	-	-	na	none, treated with denosumab	never operated on and in remission	na	96	awd
10	F	14	Radius, distal epiphysis	na	na	+	-	-	na	curettage and bone grafts	NO	none	120	ned
11	F	15	T12 vertebra	na	na	-	-	-	+	vertebrectomy and reconstruction	NO	infection	180	ned
12	F	14	Tibia, proximal meta-epiphysis	closed	5	+	-	-	na	curettage and cement	NO	none	lost	na
13	F	13	Tibia, proximal eccentric, meta-epiphysis	closed	4	+	-	-	na	curettage and cement	NO	none	12	ned
14	F	14	Distal femur, pathological fracture	na	na	+	-	-	na	resection and prosthesis	NO	none	144	ned
15	M	15	Tibia, distal, eccentric, meta-epiphysis	closed	6	+	-	-	na	curettage and cement	NO	none	6	ned

Legend: M: male; F: female; IHC: immunohistochemistry; LR: local recurrences; FU: follow-up; na: not available; ned: no evidence of disease; awd: alive with disease.

**Table 2 cancers-13-02585-t002:** Summary of the principal case series review of English literature of GCTB in pediatric patients.

Publication	Range of Time Cases	Sex Age	Bone	Status of Growth Plate	Location	Treatment	Note	Molecular Analysis
Ajay Puri et al., 2007 [7]	January 2000 to December 2005 17 patients	14 F (82%) 3 M (18%) 10–18 years	lower end of the femur (*n* = 5, 29%)	all open (17–100%)	13 (76.5%) epiphysiometaphyseal in location	14 IIC intralesional curettage	2 local recurrences	
			the upper end of the tibia (*n* = 4, 24%)					
			the upper end of the fibula (*n* = 2)		2 lower end-radius			
			distal end of radius (*n* = 2)		2 not applicable	3 wide excisions	1 pulmonary nodule	
			patient each of the upper end of the humerus, metacarpal, clavicle, and cuboid (*n* = 1)					
ThaleM. Asp Strøm et al., 2016 [6]	1984 to 2015 16 patients	12 F (75%) 4 M (25%) 6–15 years	tibia (*n* = 4, 25%)	all open (16–100%)	4 (25%) epiphysiometaphyseal distal	15 curettage 1 excision	2 local recurrences	
			fibula (*n* = 3, 18.75%)		3 (18.75%) epiphysiometaphyseal proximal			
			clavicula (*n* = 3, 18.75%)		1 (6.25%) proximal epiphysis			
			III metatarsal (*n* = 2, 12.5%)		3 (18.75%) proximal (short bones)		1 multicentric disease	
			sacrum (*n* = 2, 12.5%)					
			scapula (*n* = 1, 6.25%)		2 (12.5%) distal (short bones)			
			radius (*n* = 1, 6.25%)					
David C. Dahli et al., 1969 [5]	1910 to 1969 21 patients (75%) 7 patients (25%)	21 F (75%) 15–20 years		uk			no malignant transformation	
		7 M (75%) 12–14 years						
Alyaa Al-Ibraheemi et al., 2016 * [4]	all curettage and resection; all specimens of primary “GCT of bone” from patients 18 years old or younger	43 F (68%) 20 M (32%) 8–18 years	tibia (*n* = 16, 25%)	radiologic images (*n* = 15): 7 patients with open physes (47%); 8 patients with closed growth plates (53%)	7 (21%) cases involved the metaphysis without extension into the epiphysis	curettage and resection	1 multifocal	4 patients G34W 1 patient G34L
			femur (*n* = 14, 22%)				21 local recurrences (38%)	
			vertebral body (*n* = 13, 21%)		23 (70%) cases involved the epiphysis and metaphysis		2 (4%) pulmonary metastases 15 and 20 months after the diagnosis	
			radius (*n* = 4, 6%)					
			humerus (*n* = 4, 6%)					
			metacarp (*n* = 3, 5%)					
			fibula (*n* = 2, 3%)				5-year progression-free survival was observed in 57% (95% confidence interval, 43–71%)	
			patella (*n* = 2, 3%)					
			calcaneus (*n* = 1, 2%)					
			navicular (*n* = 1, 2%)					
			phalanx (*n* = 1, 2%)		3 (9%) cases were confined to the epiphysis			
			pelvis (*n* = 1, 2%)					
			ulna (*n* = 1, 2%)					
Carmen Sydlik et al., 2020 [8]	children underwent therapy with denosumab between September 2011 and December 2014 4 patients	1 F (25%) 3 M (75%) 6–13 years	solid variant of ABC in the left os sacrum (*n* = 1, 25%)	uk		children with severe hypercalcemia after treatment with denosumab for unresectable giant cell tumors of bone and for aneurysmal bone cysts	1 patient developed pulmonary metastasis	
			a giant cell tumor in lumbosacral spine (L5/S1) (*n* = 1, 25%)					
			left thigh and aneurismal					
			bone cyst with typical osteoclast-like giant cells and intense vascularization (n = 1, 25%)					
			a giant cell tumor localized in Th2					
Picci Piero et al., 1983 [9]	giant-cell tumor of bone in skeletally immature patients 6 patients	5 F (90%) 1 M (10%) 10–14 years	proximal fibula (*n* = 1; 10%)	6 patients with open physes (100%)	epiphyseal plate involvement (*n* = 5/6; 83%)	marginal resection (*n* = 1; 10%)		
			distal femur (*n* = 4; 80%)			wide resection (*n* = 2; 40%)		
			proximal tibia (*n* = 1; 10%)			curettage (*n* = 3; 60%)		

* Recurrent and metastatic tumors.

## Data Availability

Data available on request due to restrictions (e.g., privacy or ethics). The data presented in this study are available on request from the corresponding author.

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
