# Peer review of "Giant Cell Tumor of Bone in Patients under 16 Years Old: A Single-Institution Case Series"

_cancers, 2021, doi:10.3390/cancers13112585_

Round 1

Reviewer 1 Report

Major comment

1.- It should be discussed in a deeply way the reasons of potential false negative in sequencing analysis, and why other components of the immunohistochemistry of GCTB (RANK, DC33, CD14) are not specifically enough for diagnostic confirmation.

Minor comments

1.- In the abstract, authors state as less than 140 the frequency of GCTB in younger than 16, and in the introduction this figure is less than 130. Please modify for a convenient consistency.

2.- “confirming previous reports” instead of “report” in the Discussion section.

3.- The appearance of table 2 should be improved. It might be desirable to relocate some information into the text.

4,. How the cases have been selected? Based on pathology report? Based on radiological reports? Based on the registry of sarcoma board?

5.- Comments on the potential disturbing of decalcification processes in these results?

Author Response

Major comment

1.- It should be discussed in a deeply way the reasons of potential false negative in sequencing analysis, and why other components of the immunohistochemistry of GCTB (RANK, DC33, CD14) are not specifically enough for diagnostic confirmation.

Thank you for your kind revision and observation. According to your suggestions, we added an additional test to rule out the potential false negative in sequencing analysis. Considering that Sanger sequencing, the most used method with a limitation due to its limit of detection of about 20%;  to improve sensitivity, the 3 negative cases were also tested with Allele Specific Locked Nucleic Acid quantitative PCR (ASLNAqPCR) as we described previously (new ref. n. 25). ASLNAqPCR appears to be more sensitive as it identifies mutations resulted false negative when analyzed by Sanger sequencing. Since ASLNAqPCR results confirmed Sanger sequencing, the 3 negative/wilde type case were excluded from the study. We added some clarification in the text (line 110-111, 119-122, 139-141) and we cited 3 additional papers (reference n. 23. 25. 26.). Regarding RANK, CD33, and CD14, we did not use in present study, since in our experience these antibodies do not show a sufficient diagnostic accuracy; moreover, in the last WHO classification, they are not considered for the diagnosis of GCTB.

Minor comments

1.- In the abstract, authors state as less than 140 the frequency of GCTB in younger than 16, and in the introduction this figure is less than 130. Please modify for a convenient consistency.

Thank you for your comment. As you suggested we corrected the total number of GCTB in the pediatric population as less than 140 in line 44).

2.- “confirming previous reports” instead of “report” in the Discussion section.

Thank you for your suggestion; we modified the text according to your suggestions (line 204).

3.- The appearance of table 2 should be improved. It might be desirable to relocate some information into the text.

Thank you for your suggestion. A part of the information in Table 2 is now relocated in the text; we also added a new column “status of growth plate”.

4.- How the cases have been selected? Based on pathology report? Based on radiological reports? Based on the registry of sarcoma board?

Thank you for your comment and interest. All cases considered for the present study have been selected from our digital pathological archives based on pathological reports. Afterward, all clinical and radiological information was retrieved from medical records.

5.- Comments on the potential disturbing of decalcification processes in these results?

Thank you for your comment. According to your suggestions, we specified that all immunohistochemical investigations have been performed on selected tissue that was not decalcified (line 95) added in the text explains this aspect.

Reviewer 2 Report

Reference 9 cited from JBJS was in 1983 and your paper reviewed all cases of pediatric GCT from 1982-current. Since both manuscripts come from IOR, were any cases in both series?

Since there is some histologic overlap between solid ABC and GCT, were any of the cases of solid ABC or G34w negative GCTs tested for USP6 rearrangements?

Tables 1 and 2 are a bit clumsy and confusing with respect to informational content. It would be relevant and highly informational if one of the columns in each could be "status of growth plate" or "status of physics" and you could choose either "open, closed, or "indeterminate." Your descriptions and data suggest that in only two of 15 pediatric patients had open phases and that in both of those the lesions were pretty much confined to the metaphases. Since usual GCT is described as a lesion that extends to the end of the bone (but has a metaphysical component) and usual patients with GCT are skeletally mature, that would mean that one would not expect GCT to extend to the bone end in the presence of an open growth plate (something that was always emphasized by Dr. Campanacci). It would be useful to expound upon this. 

Some of your English usage needs to be altered, but in general it is good--for instance, "pronounced female predominance" is much better than "very sharp female predominance" and "with a not clear cut" should be "without clear cut."

While acronyms are ok to use in a paper, it is possible that some of your readers may not understand them if at least the first time they are introduced they are spelled out.  For example, the first time it is introduced, it should say Aneurysmal Bone Cyst (ABC).

On page 2, line 51-52, the last sentence should be altered a bit. You are equating giant cell lesions arising in Paget's with true Giant Cell Tumors, and most authors don't consider these true giant cell tumors (in fact, I don't think anyone has ever found G34W positivity in one of these lesions). I would change the sentence to something like... "In addition, giant cell rich lesions histologically similar to giant cell tumors of bone are observed in a specific subset of patients with Paget's disease" and use some better references than the ones you have chosen.

Author Response

1.- Reference 9 cited from JBJS was in 1983 and your paper reviewed all cases of pediatric GCT from 1982-current. Since both manuscripts come from IOR, were any cases in both series?

Thank you for your interest and improving comment. The paper is written by Picci et al. in 1983 presented cases treated in our Institute before 1982, and for this reason they are not considered in the present paper; we specified this in line 185-187. We selected cases from 1982 since from this year we generally have available clinical and radiological information of our cases.

2.- Since there is some histologic overlap between solid ABC and GCT, were any of the cases of solid ABC or G34w negative GCTs tested for USP6 rearrangements?

Thank you for your interest and comment. For the present study, we selected a control group of solid ABC with morphological overlaps and problems in the differential diagnosis with GCTB. All these cases were tested with immunohistochemistry and genetic for H3F3A and were negative, thus confirming the diagnosis of solid ABC, also considering the radiological features. Only one case with doubtful radiological features was also genetically tested for USP6 and was positive.

3.- Tables 1 and 2 are a bit clumsy and confusing with respect to informational content. It would be relevant and highly informational if one of the columns in each could be "status of growth plate" or "status of physics" and you could choose either "open, closed, or "indeterminate." Your descriptions and data suggest that in only two of 15 pediatric patients had open phases and that in both of those the lesions were pretty much confined to the metaphases. Since usual GCT is described as a lesion that extends to the end of the bone (but has a metaphysical component) and usual patients with GCT are skeletally mature, that would mean that one would not expect GCT to extend to the bone end in the presence of an open growth plate (something that was always emphasized by Dr. Campanacci). It would be useful to expound upon this.

Thank you for your accurate and improving comments. According to your suggestion, for our study population, a column "status of growth plate" 1 was added in Table 1. An identical column was added in Table 2. We also discussed better this problem in the Results (line 150-154), Discussion (line 207-223), and Conclusions (line 242-245). The final massage is that in skeletally immature patients with an open growth plate, GCTB can be limited to the metaphisis or have a predominant metaphyseal location, with an extension into the epiphysis, since an open growth plate did not seem to prevent GCT from penetrating the epiphyseal cartilage.

4.- Some of your English usage needs to be altered, but in general it is good--for instance, "pronounced female predominance" is much better than "pronounced female predominance" and "with a not clear cut" should be "without clear cut."

Thank you for your suggestion; we checked the English spelling in the paper.

5.- While acronyms are ok to use in a paper, it is possible that some of your readers may not understand them if at least the first time they are introduced they are spelled out.  For example, the first time it is introduced, it should say Aneurysmal Bone Cyst (ABC).

Thank you for your suggestion; we checked all the acronyms in the paper.

6.-On page 2, line 51-52, the last sentence should be altered a bit. You are equating giant cell lesions arising in Paget's with true Giant Cell Tumors, and most authors don't consider these true giant cell tumors (in fact, I don't think anyone has ever found G34W positivity in one of these lesions). I would change the sentence to something like... "In addition, giant cell rich lesions histologically similar to giant cell tumors of bone are observed in a specific subset of patients with Paget's disease" and use some better references than the ones you have chosen.

Thank you for your interesting observation. The sentence was modified according to your suggestion; moreover, we added a new reference (n. 13).